# *SLC6A4* DNA Methylation Levels and Serum Kynurenine/Tryptophan Ratio in Eating Disorders: A Possible Link with Psychopathological Traits?

**DOI:** 10.3390/nu15020406

**Published:** 2023-01-13

**Authors:** Marica Franzago, Elena Orecchini, Annamaria Porreca, Giada Mondanelli, Ciriana Orabona, Laura Dalla Ragione, Marta Di Nicola, Liborio Stuppia, Ester Vitacolonna, Tommaso Beccari, Maria Rachele Ceccarini

**Affiliations:** 1Department of Medicine and Aging, School of Medicine and Health Sciences, “G. d’Annunzio” University, 66100 Chieti, Italy; 2Center for Advanced Studies and Technology, “G. d’Annunzio” University, 66100 Chieti, Italy; 3Department of Medicine and Surgery, University of Perugia, 06129 Perugia, Italy; 4Laboratory of Biostatistics, Department of Medical, Oral and Biotechnological Sciences, “G. d’Annunzio” University, 66100 Chieti, Italy; 5Food Science and Human Nutrition Unit, University Campus Biomedico of Rome, 00128 Rome, Italy; 6Department of Psychological, Health and Territorial Sciences, School of Medicine and Health Sciences, “G. d’Annunzio” University, 66100 Chieti, Italy; 7Department of Pharmaceutical Sciences, University of Perugia, 06126 Perugia, Italy

**Keywords:** eating disorders, anorexia nervosa, bulimia nervosa, binge eating, serotonin, DNA methylation, *SLC6A4*, epigenetics, kynurenines, tryptophan

## Abstract

Background: The incidence of eating disorders (EDs), serious mental and physical conditions characterized by a disturbance in eating or eating-related behaviors, has increased steadily. The present study aims to develop insights into the pathophysiology of EDs, spanning over biochemical, epigenetic, psychopathological, and clinical data. In particular, we focused our attention on the relationship between (i) DNA methylation profiles at promoter-associated CpG sites of the *SCL6A4* gene, (ii) serum kynurenine/tryptophan levels and ratio (Kyn/Trp), and (iii) psychopathological traits in a cohort of ED patients. Among these, 45 patients were affected by restricting anorexia nervosa (AN0), 21 by purging AN (AN1), 21 by bulimia (BN), 31 by binge eating disorders (BED), 23 by unspecified feeding or eating disorders (UFED), and finally 14 by other specified eating disorders (OSFED) were compared to 34 healthy controls (CTRs). Results: Kyn level was higher in BED, UFED, and OSFED compared to CTRs (*p* ≤ 0.001). On the other hand, AN0, AN1, and BN patients showed significatively lower Kyn levels compared to the other three ED groups but were closed to CTRs. Trp was significantly higher in AN0, AN1, and BN in comparison to other ED groups. Moreover, AN1 and BN showed more relevant Trp levels than CTRs (*p* <0.001). BED patients showed a lower Trp as compared with CTRs (*p* ≤ 0.001). In addition, Kyn/Trp ratio was lower in the AN1 subtype but higher in BED, UFED, and OSFED patients than in CTRs (*p* ≤ 0.001). SCL6A4 DNA methylation level at CpG5 was lower in AN0 compared to BED (*p* = 0.021), and the CpG6 methylation was also significantly lower in AN0 in comparison to CTRs (*p* = 0.025). The mean methylation levels of the six CpGs analyzed were lower only in the AN0 subgroup compared to CTRs (*p* = 0.008). Relevant psychological trait EDI-3 subscales were correlated with biochemical and epigenetic data. Conclusions: These findings underline the complexity of psychological and pathophysiological components of EDs.

## 1. Introduction

Eating disorders (EDs) are serious mental and physical conditions characterized by a disturbance in eating or eating-related behaviors and are correlated with distressing thoughts and emotions [1]. EDs are heterogeneous and are usually associated with physical and mental health morbidity and elevated mortality, which vary according to the diagnosis, duration, and frequency of certain specific behaviors [2,3,4]. Comorbidities with other psychiatric diseases are high [5] since up to 95% of EDs subjects have at least one additional psychiatric illness [6]. EDs are prevalent among females and young adults, with a female-to-male ratio ranging from 3:1 to 8:1 [7,8]. Based on the *Diagnostic and Statistical Manual of Mental Disorders* (DSM-V) [1], EDs include anorexia nervosa (AN), bulimia nervosa (BN), binge eating disorder (BED), other specified eating disorders (OSFED), unspecified feeding or eating disorder (UFED), pica, rumination disorder, avoidant/restrictive food intake disorder (ARFID), and others.

The pathogenesis of EDs ensues from a pattern of genetic, psychosocial, biological, and environmental risk factors [9,10], and its complexity is often increased by the shift from one to another eating disorder diagnosis [6,11] (typically within the first 5 years of illness), suggesting an overlapping pathogenesis of this type of disorder [12]. In particular, it has been demonstrated that EDs share common neurobiological abnormalities, such as dysregulation of the serotonergic system [13]. Serotonin (5-hydroxytryptamine, 5-HT) is a monoamine that acts as a neurotransmitter. Only about 3% of the essential amino acid L-tryptophan (Trp), purchased from the diet, is conveyed to the synthesis of 5-HT, which is involved in behavioral, biological, and physiological functions [14] regulating sleep, mood, drug abuse, appetite, and eating behavior [15]. Some studies have suggested that underweight AN patients have reduced 5-HT levels [16,17] as a consequence of reduced Trp availability probably due to the hypocaloric diet. However, after refeeding, peripheral measures of 5-HT tend to normalize [18]. On the contrary, the serum Trp concentration was very high in AN subjects with more severe statuses (lower BMI, excessive exercise, and low energy intake), probably correlated with measures of catabolism [19]. A decrease in 5-HT is also associated with anxiety, depression, autism spectrum disorder, and obesity [20].

Several hypotheses have been suggested to explain the serotoninergic tone alteration in the brain of ED patients, including the observation of dysfunctional 5-HT receptors (e.g., 5-HT2AR) and 5-HT transporters. Specifically, the serotonin transporter (SERT, 5-HTT), an integral membrane protein abundantly expressed in the brain, plays a key role in the regulation of 5-HT homeostasis and was thought to be implicated in EDs [21] by controlling hunger and promoting satiety. SERT is encoded by a gene (also known as the solute carrier family 6 member 4, *SLC6A4*) located on chromosome 17q11.2 [22]. Genetic variants within the promoter of this gene (i.e., rs25531 and rs25532) along with variable number tandem repeat (VNTR) polymorphisms in the promoter and intron 2 regions (5HTTLPR and STin2) modulate its transcriptional activity [23], influencing emotional, endocrine, and personality characteristics as well as vulnerability to a broad range of psychiatric disorders [24,25,26,27]. In this scenario, in vivo study demonstrated that transgenic mice overexpressing *SLC6A4* are lighter and shorter than controls [28]; on the other hand, *Slc6a4* knockout mice develop late-onset obesity, hepatic steatosis, glucose intolerance, and insulin resistance [29].

In addition to genetic variants, recent epigenetic alterations modulating gene expression have garnered much interest in the etiology of EDs [17,30], although the role of these mechanisms is currently unclear. Epigenetic modifications mainly refer to chemical modifications of DNA and chromatin proteins and are mediated by several environmental factors, such as diet and stress [31]. When epigenetic modifications occur, gene expression changes without altering DNA sequences. DNA methylation, characterized by the addition of a methyl group to cytosine residues (5 mC) in CpG dinucleotides, is the most widely studied epigenetic modification. DNA methylation of the *SLC6A4* promoter region has been associated with 5-HT-related disorders, including abnormal eating behaviors (including food restriction and food avoidance), psychiatric pathologies, depressed mood, trauma, and work stress [17,32,33,34]. *SLC6A4* promoter hypermethylation is significantly associated with an increased prevalence of obesity [35]. Due to the regulatory network of *SLC6A4*, further insight into DNA methylation could be of high interest to better understand and treat different ED subgroups.

The essential amino acid Trp is a building block for protein synthesis and the precursor of different metabolic pathways. In eukaryotic cells, Trp is conveyed in two main metabolic pathways, generating serotonin and melatonin (namely, the serotonin pathway) and kynurenines (namely, the kynurenine pathway). Trp availability is known to influence sensitivity to mood disorders [36,37]. The kynurenine (Kyn) pathway represents a major route for the metabolism of Trp in both the peripheral and central systems, in which about 90% of Trp is catabolized to form nicotine amides and the vitamin niacin [38]. The enzyme indoleamine-(2,3)-dioxygenase 1 (IDO1) controls the rate-limiting step in the conversion of Trp into the intermediate catabolite l-Kyn [39]. An altered Kyn/Trp, as biomarker of IDO1 activity, has been shown in many conditions—including Alzheimer’s disease, amyotrophic lateral sclerosis, AIDS dementia complex, cancer, depression, and schizophrenia—suggesting an important role of the kynurenine pathway in the development of several diseases [40]. Nevertheless, the link between the Kyn pathway and EDs remains poorly understood.

Through an interdisciplinary approach, the present study describes the relationship among epigenetic data (i.e, DNA methylation at the promoter-associated CpG sites of the *SCL6A4* gene), biochemical data (i.e., the measurement of serum Kyn/Trp ratio), and psychopathological traits in ED patients.

## 2. Materials and Methods

### 2.1. Participants and Healthy Control Recruitment

EDs patients were referred to the outpatient clinic for EDs at the Palazzo Francisci (USL 1 Umbria, Todi, Italy). All ED patients were unrelated and were of Caucasian origin. They were recruited consecutively at the Palazzo Francisci residence from January 2018 to July 2019. They were diagnosed according to the DSM-V (*Diagnostic and Statistical Manual of Mental Disorders, 5th edition*) criteria by at least one health professional or psychiatrist. Psychopathological features were evaluated during the first week after the admission to the residential facility together with blood collections. EDs were divided into six major groups: 66 patients were affected by AN, 21 were affected by BN (mean age 19 [17.0;26.0]), 31 by BED (mean age 39 [20.2;52.8]), 23 by UFED (mean age 54.0 [18.5;64.5]) and 14 by OSFED (mean age 54.5 [35.2;58.5]). In addition, according to DSM-V criteria, 66 AN patients were divided in turn into 45 restricting AN (AN0) (mean age 16.0 [14.0;18.0]) and 21 purging AN (AN1) (mean age 18.0 [17.0;19.0]), based on the presence or absence of binge eating and vomiting behaviors and according to anamnesis and a psychiatric test. In addition, 34 Caucasian healthy controls (CTRs) without a lifetime history of EDs, psychiatric disorders, or substance abuse were consecutively recruited in Palazzo Francisci.

### 2.2. Ethical Statement and Inclusion Criteria

The study and all research processes were approved by the Ethics Committee of the Aziende Sanitarie (CEAS) della Regione Umbria, Italy. Written informed consent was obtained from all subjects in accordance with the Declaration of Helsinki and its subsequent revisions. For data protection and confidentiality, all participants were assigned a unique research identifier alphanumeric code.

Sociodemographic characteristics and psychopathological and clinical data were collected for ED patients during the first week at the facility residence. During recruitment, all the subjects filled out a questionnaire with different queries about gender, age, BMI, secondary amenorrhea, food restriction, water restriction, fasting, diet pills, binge eating, vomiting, alcohol abuse, laxative abuse, diuretics abuse, drug abuse, excessive physical activity, and sleep disorder. In the survey, the criterion for food restriction to standardize all answers was: eating <1000 kcal/day, and patients could answer Yes or No. Water restriction was indicated in the questionnaire as drinking <0.5 L/day (Yes or No).

Exclusion criteria for the admission, determined before psychological test [41], included mental retardation, dementia, schizophrenia, Turner’s syndrome, other neurological disorders, and underlying endocrine pathologies. CTRs inclusion criteria include never having been diagnosed with any previous EDs and/or familiarity with EDs, any psychiatric disorders and/or familiarity, and having normal BMI for their whole life (range 18.5–24.9 kg/m^2^).

Before starting the rehabilitation course in the residential structure Palazzo Francisci, all patients carried out a complete blood test.

### 2.3. Psychological Traits

Psychological profiles were analyzed through the Italian version of the self-completed *Eating Disorders Inventory, third edition* (EDI-3) [42]. EDI-3 measures psychological traits relevant in individuals with Eds, characterizes them by 91 items, and organizes them into 3 eating-disorder-specific scales and 9 general psychological scales. It also generates six composites: eating disorder risk (EDRC), ineffectiveness (IC), interpersonal problems (IPC), affective problems (AP), overcontrol (OC), and global psychological maladjustment (GPM). 

### 2.4. DNA Methylation Analysis

A blood sample was collected in a sterile tube containing EDTA for all patients, and CTRs and has been used for DNA extraction and serum analysis.

Genomic DNA was extracted from peripheral blood lymphocytes starting from 300 μL of whole anticoagulated peripheral blood using a commercially gDNA Mini Kit (Geneaid) according to the manufacturer’s instructions. An amount of 100 ng of each sample was run in 0.8% agarose gel electrophoresis in 1X TBE buffer at 90 V for 45 min to verify the integrity of gDNA, and at least the gDNA was quantified using the NanoDrop system [43].

DNA methylation levels at 6 CpG sites within the *SLC6A4* promoter region (ENSG00000108576, Chr:17) were assessed using pyrosequencing (Qiagen, Hilden, Germany, Pyromark Q96) (Figure 1). In brief, genomic DNA was treated with sodium bisulfite (NaBis) using the EpiTect Plus DNA Bisulfite Kit (Qiagen) as previously reported [44]. Bisulfite-treated DNA (50 ng) was amplified via PCR using the Pyromark PCR kit (200) (Qiagen) with a biotin-labeled primer (Hs_SLC6A4_01_PM PyroMark CpG assay PM00065625). The PCR cycling conditions according to the general guidelines of pyrosequencing were as follows: 95 °C for 15 min, 45 cycles at 94 °C for 30 s, 60 °C for 30 s, 72 °C for 30 s, and a final extension at 72 °C for 10 min. DNA methylation levels were analyzed through the PyroMark Q96 ID version 1.0.11 software, which calculates the methylation percentage expressed both as a percentage of every single CpG site and as the average of the methylation percentage of all 6 of the CpG sites under study.

### 2.5. HPLC Analysis

Plasma levels of Trp and its derived metabolite Kyn, as well as the Kyn/Trp ratio, were evaluated at the admission of ED patients. Blood samples were immediately centrifuged to obtain serum samples. After deproteinization, Kyn and Trp concentrations were measured via HPLC. Briefly, a Perkin Elmer (series 200 HPLC) instrument and a Kinetex C18 column (250 × 4.6 mm, 5 µm, 100 Å, Phenomenex, Torrance, CA, USA) were used for the analysis. The column was maintained at a temperature of 25 °C and a pressure of 1800 PSI. Samples were eluted with a phase containing 10 mM NaH_2_PO_4_ (pH = 3.0; 99%) and methanol (1%) (Sigma–Aldrich, St. Louis, MO, USA) with a flow rate of 1.3 mL/min. Kyn was detected at 360 nm and Trp was detected at 220 nm by UV detector. The software TotalChrom version 6.3.1 was used for evaluating the concentration of Kyn and Trp in samples using a calibration curve.

### 2.6. Statistical Analysis

The descriptive analysis reports, in accordance with the distribution of the variables, the median (q1 = first quartile; q3 = third quartile) for continuous variables and the absolute frequency and percentage for categorical variables. Pearson chi-square test was used to investigate the association among categorical variables. The Kruskal–Wallis test was used to assess whether statistically significant differences existed between the diagnosis groups for the continuous variables under study, and if a *p* ≤ 0.05 was found, Dunn’s post hoc test was applied. Mann U Whitney test was used to test unpaired differences between groups. Nonparametric Spearman rho correlation coefficients were estimated to evaluate the relationship between quantitative variables. Because of the exploratory nature of the study, no correction for multiplicity was applied. Statistical analysis was performed using the R environment for statistical computing and graphics version 3.5.3 (R Foundation for Statistical Computing, Vienna, Austria).

## 3. Results

### 3.1. Clinical Data

Sociodemographic characteristics and clinical phenotypes are shown in Table 1 and Table 2, respectively. Women were prevalent in patients with EDs with the following frequencies: 100% in both AN0 and AN1, 95.2% in BN, 74.2% in BED, 82.6% in UFED, and 85.7% in OSFED (Table 1). The BED group contains the highest percentage of males (25.8%) of all groups. Notably, among patients with EDs, at least one out of three patients declared that a first-degree family member had had a previous ED. As expected, AN patients showed lower weight than the other groups (Table 1), with a BMI outflank of 14.4 kg/m^2^ for AN0 and 15.8 kg/m^2^ for AN1. This justifies a severe or extreme severe condition based on BMI value as described in DSM-V. At the same time, some patients with BN showed a BMI under normality (with a median = 17.7 kg/m^2^) probably due to a possible diagnostic crossover from AN purging subtype and BN during the life history of the disease. On the other hand, severe obesity was registered in the BED, UFED, and OSFED groups, with a maximum BMI in the BED subgroup (median 45.7 kg/m^2^), followed by UFED (42.3 kg/m^2^) and lastly OSFED (38.9 kg/m^2^). It is worthy of note that the 65.5% and 65.2%, respectively in BED and UFED groups presented a grade III obesity.

As described in Table 2, secondary amenorrhea was prevalent in AN0 (89.5%), followed by AN1 and BN (63.2% and 62.5%). Menopause was absent in AN0, AN1, and BN because of the young age, whereas it was present in BED, UFED, and OSFED in a different way (42.1, 43.8, and 20%, respectively). Food restriction was a behavioral characteristic present in almost all AN1 and AN0 (100%), in a large group of BN (94.4%), and in more than half of the other three groups (>60%).

When we refer to dietary restriction behavior, we refer to qualitative and quantitative behavior: eating fewer meals per day, higher frequency of fasting, consuming small meals, and only low-calorie foods. Water restriction was a condition almost present in extreme patients (54.5% of AN0, 52.4% of AN1, 33.3% of BN, 16.0% of BED, 10% of UFED, and 8.33% of OSFED). Patients, in particular those affected by AN, decided to reduce water intake due to incessant worry over gaining weight. Sometimes, all biological functions are impaired and patients lose the sense of thirst. In the survey, fasting common in AN and BN (around 80%) is defined as skipping at least one meal/day. Binge eating was infrequent in patients with AN0 but was a common trait for all other subgroups, with an incidence of 55.0% in AN1, 88.9% in BN, 100% in BED, 60.0% in UFED, and 83.3% in OSFED (*p* < 0.001). Inappropriate compensatory behavior (self-induced vomiting) was a common trait in AN1 and BN, as reported in Table 2: 85.0% and 88.9%, respectively (*p* < 0.001). Alcohol abuse was unusual; only one out of four patients with BN declared to have alcoholism. Around 30% of AN1 and BN patients stated that they engaged in laxative abuse, whereas all other patients did so less (Table 2). Diuretics and drug abuse were unusual in all ED subgroups (Table 2). The excessive physical activity is significantly linked (*p* < 0.001) to a peculiar portion of patients, as reported in Table 2: AN0 (79.5%), AN1 (90.0%), and BN (66.7%). On the contrary, it is undistinguished in BED (7.7%), UFED (10.0%), and OSFED (16.7%). Finally, sleep disorder was present in around 80% of AN0, AN1, and OSFED patients but only in half of BN, BED, and UFED. Only these last two subtypes used a CPAP (continuous positive airway pressure) (*p* = 0.002).

Glucose level was also statistically different among EDs groups, albeit all values were in the range of normality (70.0–100.0 mg/dL). AN patients (purging and restricted subtypes) presented lower fasting glucose levels in line with their malnutrition status.

#### Hematological Data

Malnutrition was evident in AN0 and AN1. White blood cell (WBC) count and platelet (PLT) count decreased significantly in AN0 and AN1 patients and partially in BN patients. Despite a decrease in WBC count, the patients did not present any infections. Red blood cell (RBC) count was borderline in AN0, AN1, and BN, suggesting the onset of anemia (Table 3).

### 3.2. Biochemical Data

Kyn levels, as reported in Table 4, were higher in BED, UFED, and OSFED than in CTRs (*p* ≤ 0.001). On the other hand, purging and restricting AN as well as BN patients showed significantly lower Kyn levels compared to other ED groups but similar levels to CTRs. Trp was significantly higher in AN0, AN1, BN in comparison with other EDs groups. Moreover, AN1 and BN with a Trp median value upper than 50 µmol/L showed relevant results if compared to CTRs (*p* < 0.001). Conversely, BED patients with a median of 41.0 µmol/L showed a lower Trp as compared with CTRs (*p* < 0.001). In addition, the serum Kyn/Trp ratio was significantly lower in the AN1 subgroup but significantly higher in BED, UFED, and OSFED patients, compared to the CTRs (*p* < 0.001). AN0, AN1, and BN showed significantly lower Kyn/Trp ratios compared to other ED groups and stood out for severe obesity.

In the whole cohort of ED patients, several interesting correlations were observed between plasma levels of Kyn and Trp, and the Kyn/Trp ratio was correlated with peculiar behaviors of ED patients, such as vomiting, excessive levels of physical activity, and binge eating behavior (Figure 2). Specifically, vomit (characteristic in AN1 and BN subtypes) and excessive levels of physical activity (present in AN0 and AN1 and in some BN patients) were both positively correlated with a lower Kyn level and higher Trp value. As a consequence, the resulting Kyn/Trp ratio was very low in patients that declared that they vomited (*p* = 0.001) or abused physical activity (*p* < 0.001) compared to all other patients. On the contrary, in patients with binge eating behavior, the Kyn/Trp ratio resulted increased (*p* < 0.001) as a consequence of increased Kyn and a drastical drop in Trp.

### 3.3. Epigenetic Data

*SCL6A4* DNA methylation level at CpG5 was lower in AN with restricting subtype (AN0) compared to BED (*p* = 0.021), and the CpG6 methylation was significatively lower in AN0 in comparison to CTRs (*p* = 0.025). The mean methylation levels of the six CpGs analyzed were lower in AN0 compared to CTRs (*p* = 0.008) (Table 4).

Moreover, DNA methylation at CpG4 was positively related to Kyn concentration (rho = 0.764; *p* < 0.001) as well as to Kyn/Trp ratio (rho = 0.809; *p* < 0.001) in the OSFED group. DNA methylation at CpG1 and CpG6 in AN0 was positively correlated with Kyn/Trp ratio (rho = 0.418; *p* < 0.05 and rho = 0.428; *p* < 0.05, respectively). Indeed, CpG1 was positively correlated with Kyn concentration (rho = 0.493; *p* < 0.05) in the same group. In addition, DNA methylation at CpG5 and CpG6 was also positively related to Trp (rho = 0.459; *p* < 0.05) and Kyn (rho = 0.471; *p* < 0.05) in the BED group, respectively (Appendix A).

### 3.4. Psychopathological Data

Psychopathological data for ED patients are reported in Table 5. Significant differences between the EDs groups were found on the subscales of the EDI-3, covering a wide range of ED symptoms and pathology. In addition, EDI-3 subscale scores were correlated with biochemical and epigenetics data. *SLC6A4* methylation levels were inversely related to relevant psychological trait subscales, including low self-esteem (LSE) (rho = −0.435; *p* < 0.05 vs. CpG2), interpersonal alienation (IA) (rho = −0.493; *p* < 0.05 vs. CpG4), and ineffectiveness composite (IC) (rho = −0.417; *p* < 0.05 vs. CpG2) in the AN0 group. In addition, interoceptive deficits (ID) were inversely related to DNA methylation at CpG1 (rho = −0.649; *p* < 0.05) and CpG2 (rho = −0.635; *p* < 0.05). Indeed, DNA methylation at CpG4 was positively related to interpersonal alienation (IA) (rho = 0.636; *p* < 0.05) and emotional dysregulation (ED) (rho = 0.798; *p* < 0.01) and CpG6 with ED (rho = 0.830; *p* < 0.001) in the AN1 group. The ID (rho = −0.681; *p* < 0.05) was inversely related to Kyn levels in UFED patients.

## 4. Discussion

To the best of our knowledge, the present study investigates for the first time the complexity of EDs by an interdisciplinary approach, deepening the relationship between *SCL6A4* gene DNA methylation, serum Kyn/Trp ratio, and psychopathological traits in patients with different ED subtypes compared to the CTR group. Poor and contradictory results are available on the kynurenine pathway in EDs patients. Consistent with the observation in mice, obese patients have lower Trp and higher Kyn in plasma [45]. Breum et al. reported that plasma Trp concentrations and the ratio of Trp to other large neutral amino acids in obese subjects were low throughout a 24 h period and that these effects persisted after weight reduction [46]. The hypothesis could be that a lower Trp level observed throughout the day and night in obese subjects causes a reduced 5-HT synthesis in the brain and suggests that the obese may struggle against a biochemical signal oriented toward increased appetite and food intake [47,48]. Moreover, the kynurenine pathway is chronically activated in obesity in response to the low-grade inflammatory condition that characterizes these patients, and it has been associated with neuropsychiatric symptoms in obese patients [49]. In this scenario, we speculated that the available Trp, taken with their diets, could constantly be conveyed into the Kyn pathway in obese patients, resulting in increased Kyn levels and decreased 5-HT. Higher Kyn/Trp ratios and parallel lower 5-HT levels were also observed in obese subjects (without a specific ED diagnosis) relative to non-obese controls [50].

In our cohort, serum Trp resulted significantly higher in AN0, AN1, and BN patients, whereas BED patients showed a lower Trp level compared to CTRs. In our cohort, Kyn levels were higher in BED, UFED, and OSFED groups than in CTRs. On the other hand, AN with restricting and binge–purge subtypes as well as BN patients showed a lower Kyn level compared to other EDs groups.

In our cohort, serum Kyn/Trp ratios were higher in BED, UFED, and OSFED patients than in the CTRs. As previously described [39], Kyn and Trp levels are correlated with each other because Kyn is generated from Trp along the kynurenine pathway, which is essentially controlled by TDO and IDO1 enzymes. Kyn levels increased in patients with binge eating, probably because of the persistent inflammatory state in adipose tissue and liver activating the Trp-degrading enzymes. As a consequence, Trp level decreased in the same patients. We speculated that binge eating patients internalized from their diet highs amounts of the essential amino acid Trp, which was immediately converted into Kyn. In fact, the Kyn pathway accounts for ~95% of dietary Trp degradation, of which 90% is attributed to the hepatic synthesis. As a consequence, Kyn/Trp ratio was very high in binge eating patients [51].

For the first time, we also evaluated the serum Kyn/Trp ratio in AN (restricted and purging) and BN subtypes, and this value was lower compared to other ED groups, in particular in the purging AN group. The value was significantly below that of the CTRs.

Although altered levels of the Trp-derived metabolites along the Kyn pathway have been observed in the pathogenesis of several conditions, including aging and sleep disorders, metabolic syndrome, cardiovascular and autoimmune diseases, anxiety and depression, and neurodegenerative diseases [52], so far, poor observations are available on the alteration of Trp–Kyn pathway EDs. Food restrictions have been described to reduce available circulating Trp and therefore to induce serotoninergic system depletion [53]. Mood disorders and neuropsychiatric symptoms—such as depression, anxiety, and others—are correlated with EDs in general and in particular with AN—especially during the acute phase [54], probably due to the Trp deficiency—but the literature on the topic is contrasting. On one hand, plasma Trp significantly increased during the refeeding process in underweight AN patients [55]. Conversely, Favaro et al. (2000) demonstrated in sixteen AN patients that Trp level—in particular, the ratio between Trp and other large neutral amino acids (Trp/LNAA)—competes with Trp for brain uptake [19]. In this work, Trp/LNAA value was very high in the AN subjects with more severe statuses (lower BMI, excessive exercise, and low energy intake) and probably correlated with measures of catabolism. In the presence of low fat stores, depleted glycogen, and insufficient Kcal intake, AN patients have to turn to release amino acids from the skeletal muscle back into the plasma. In the middle of these opposite results, other studies demonstrated that plasma Trp levels did not change comparing underweight AN patients and CTRs [56]. In our cohort, higher Trp value was registered for restricting and purging AN subtypes as well as BN patients and was positively correlated with excessive levels of physical activity, corroborating the catabolism hypothesis.

Trp supplementation was proposed as a therapeutic approach for AN [57] and was aimed at increasing 5-HT brain levels and reducing depressive symptoms. Contrasting data were obtained in the AN group, with an increase in anxiety after Trp intravenous administration [58]. A dietary reduction of serum Trp was also associated with reduced anxiety among AN patients [59]. In fact, lean men respond to prolonged fasting by increasing hypothalamic 5-HT transporter availability, whereas this response is absent in men with obesity [60].

It is known that gene and environment interaction have a role in the development of EDs, which may involve epigenetic processes. In comparison to genetic variants, epigenetic contributions to EDs remain largely understudied. To date, as reviewed by Hübel et al., few studies have evaluated DNA methylation at selected candidate genes [61] mainly involved in synaptic transmission, endoplasmic reticulum stress response, growth hormone signaling, fluid balance, the cannabinoid system, dopamine transmission, stress response, and appetite regulation [62,63,64]. Due to the role of the serotonin transporter in the regulation of emotion, behavior, energy balance and appetite control, *SLC6A4* may also be a promising candidate in epigenetic analyses involving DNA methylation. In our ED cohort, the methylation profile of the *SLC6A4* gene resulted in hypomethylated sites in AN with the restrictive trait, indicating a potential activation of the gene transcription. Previously, animal studies showed the role of 5-HT in suppressing eating behaviors, promoting satiety, and producing anorexic effects [65,66]. Moreover, although *SLC6A4* methylation has been also shown to play a role in depressive disorders and schizophrenia [67], no difference in *SLC6A4* methylation levels in AN patients has been reported [68,69]. Nevertheless, the DNA methylation in this candidate gene was positively correlated with resting-state functional connectivity (rsFC) in specific brain regions, such as the dorsolateral prefrontal cortex. These data suggested that increased rsFC in the salience network mediated the link between *SLC6A4* methylation and eating disorder symptoms in AN patients [17].

The EDI-3 evaluates eating pathology and general psychological constructs of relevance to EDs. Previously, some variants in genes related to eating behavior and mental disorders have been correlated with at least one of the EDI scales and psychopathological traits [70,71,72,73]. Moreover, biochemical measures, including androgen and cholesterol levels, have shown significant correlations with several EDI scores [74,75]. In the current study, significant correlations emerged between *SLC6A4* methylation and Kyn levels with the scores of the EDI-3 for LSE, IA, ID, IC, MF, IPC, and GPM in several ED groups. In particular, interesting results were obtained with BED and OSFED subgroups and maturity fear (MF). BED, with a median score of 9.0, and OSFED, with 12.0, were both hypomethylated at CpG4 in *SLC6A4*. Methylation levels at CpG3, CpG4, and CpG6 were negatively associated with global psychological maladjustment (GPM) and OSFED patients. Not by chance, they showed the lowest score (62.5) compared to other subgroups.

A strength of our study is that this is the first attempt to characterize the major six ED subtypes; demonstrate the associations betweenDNA methylation profiles of the *SCL6A4* gene, the serum Kyn/Trp ratio, and psychopathological traits in the different EDs subgroups; and point out the complexity of the psychological and pathophysiological components of these conditions.

The present study has some limitations. First, we show the laboratory variables only in ED groups because in the control group, they were unavailable. Second, although the ED groups (total 155) can be considered a good sample size, it should be noted that the number of the controls was relatively small. In conclusion, our findings suggest that epigenetic marks at the *SLC6A4* gene, responsive to environmental stressors, might play a role in EDs and/or associated phenotypes, including anxiety and increased stress reactivity. In addition, considering the role of the serotonergic system, the study of the complex link between SLC6A4 expression and the serum Kyn/Trp ratio in EDs patients may be an area of high interest to better understand and treat 5-HT-related EDs. A lower SLC6A4 methylation level in AN with restricting subtype could be accompanied by a change in SLC6A4 mRNA expression, suggesting that altered *SLC6A4* methylation may be of functional relevance in EDs. Thus, since SLC6A4 plays a key role in regulating eating behaviors, epigenetic alterations in this gene could promote satiety, producing an anorexic effect. In this complex scenario, the significance of the link between *SLC6A4* methylation patterns and the serum Kyn/Trp ratio related to the specific pathophysiology of EDs must be clarified with the aim of assisting the clinicians in a more precise diagnosis of the various EDs. In this view, it should be also elucidated whether or not epigenetic changes occur at first stage or when illness has taken full hold. Such knowledge may pave the way toward future research into the potential of epigenetic markers to evaluate the risk for eating disorders. It is known that the interest of the epigenetic mechanisms is due to their reversibility; therefore, further investigation is needed to establish the possibility to modulate the epigenetically relevant effects through pharmacological and behavioral treatment strategies. In this view, in 2014, Domschke et al. investigated the correlation between SLC6A4 methylation level and impaired response to antidepressant drugs after six weeks of treatment in patients with major depressive disorder (MDD), suggesting that individuals’ methylation profile may contribute to improve a personalized therapy [76,77]. Epigenome-modifying drugs both up- and downregulating DNA methylation are promising and could open new avenues on EDs.

## 5. Conclusions

This study showed the associations between DNA methylation profiles of the *SCL6A4*, the serum Kyn/Trp ratio, and psychopathological traits in the different ED subgroups and pointed to the complexity of psychological and pathophysiological components of these conditions. These results provide significant insights into the pathophysiology of EDs, contributing to improve the stratification of ED patients.

## Figures and Tables

**Figure 1 nutrients-15-00406-f001:**
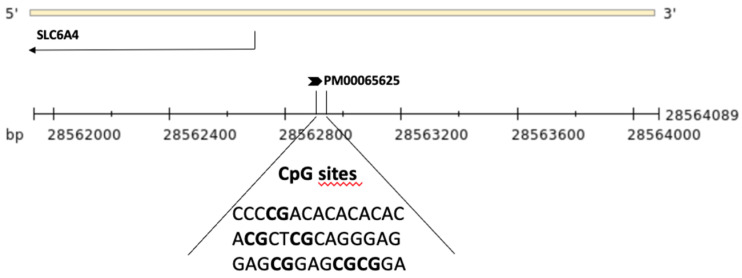
Schematic representation of the *SLC6A4* gene and the upstream region. The region sequenced to analyze DNA methylation levels of the six CpG sites (in bold) is shown.

**Figure 2 nutrients-15-00406-f002:**
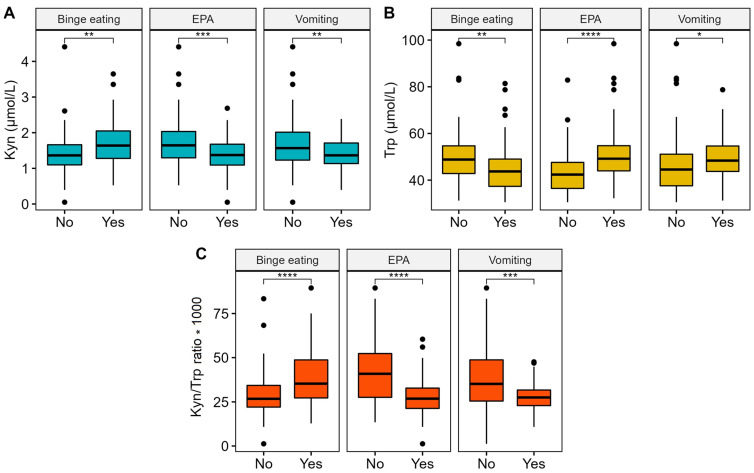
Boxplot and Mann–Whitney U test for Kyn (Panel **A**), Trp (Panel **B**), and the Kyn/Trp ratio (Panel **C**) with vomit, excessive levels of physical activity (EPA), and binge eating behavior. Significance code: * *p* < 0.05, ** *p* < 0.01, *** *p* < 0.001, **** *p* < 0.0001.

**Table 1 nutrients-15-00406-t001:** Sociodemographic and anthropometric characteristics of EDs and control group. Summary descriptive table by groups of diagnosis expressed as median (q1 = first quartile; q3 = third quartile) for continuous variable and absolute frequency (% = column percentage) for categorical. Significance values are according to the Kruskal–Wallis and chi-squared tests. Statistically significant values are in bold. * = The *p*-value refers to the difference between independent groups excluding the control group. Abbreviations: AN0 = restricting anorexia nervosa; AN1 = purging anorexia nervosa; BN = bulimia; BED = binge eating disorders; UFED = unspecified feeding or eating disorder; OSFED = other specified eating disorders; CTR = controls.

Variables	AN0	AN1	BN	BED	UFED	OSFED	CTR	*p*-Value ***
**Gender**:								** *0.001* **
Female	45 (100.0)	21 (100.0)	20 (95.2)	23 (74.2)	19 (82.6)	12 (85.7)	29 (85.3)	
Male	0 (0.0)	0 (0.0)	1 (4.76)	8 (25.8)	4 (17.4)	2 (14.3)	5 (14.7)	
**Relatives with EDs:**								*0.309*
No	27 (61.4)	17 (81.0)	12 (66.7)	17 (58.6)	13 (56.5)	6 (42.9)	34 (100.0)	
Yes	17 (38.6)	4 (19.0)	6 (33.3)	12 (41.4)	10 (43.5)	8 (57.1)	0 (0.0)	
**Age at time of admission, years**	16.0 [14.0;18.0]	18.0 [17.0;19.0]	19.0 [17.0;26.0]	39.0 [20.2;52.8]	54.0 [18.5;64.5]	54.5 [35.2;58.5]	25.0 [24.0;33.0]	** *<0.001* **
**Heigh, m**	1.62 [1.57;1.65]	1.62 [1.58;1.65]	1.67 [1.60;1.71]	1.65 [1.60;1.70]	1.60 [1.58;1.69]	1.67 [1.60;1.73]	1.70 [1.62;1.75]	*0.124*
**Minimum Weight, kg**	38.0 [34.0;40.0]	41.4 [37.6;48.0]	48.0 [45.0;55.0]	70.0 [62.2;85.5]	67.0 [58.0;85.0]	65.0 [48.0;75.0]	56.0 [53.0;60.0]	** *<0.001* **
**Maximum Weight, kg**	54.5 [50.0;58.5]	57.0 [53.0;63.0]	68.0 [60.0;75.2]	111 [106;126]	106 [96.0;128]	113 [98.5;130]	63.0 [58.5;70.5]	** *<0.001* **
**BMI, kg/m^2^**	14.4 [13.6;15.4]	15.8 [14.7;17.2]	17.7 [16.1;19.0]	26.2 [23.6;30.9]	25.4 [21.9;31.5]	22.3 [18.9;26.9]	20.5 [19.0;21.7]	** *<0.001* **
**Maximum BMI, kg/m^2^**	21.3 [19.8;23.3]	22.9 [19.7;24.6]	24.3 [23.1;25.9]	45.7 [38.3;49.3]	42.3 [37.3;46.9]	38.9 [37.1;45.9]	22.8 [21.5;23.6]	** *<0.001* **

**Table 2 nutrients-15-00406-t002:** Eating behaviors and physical characteristics of ED groups. Descriptive statistics by groups of diagnosis for eating behaviors and physical characteristics expressed as column percentage. The *p*-values results from the chi-squared test. Statistically significant values are in bold. Abbreviations: AN0 = restricting anorexia nervosa; AN1 = purging anorexia nervosa; BN = bulimia; BED= binge eating disorders; UFED = unspecified feeding or eating disorder; OSFED = other specified eating disorders, CPAP: Continuous Positive Airway Pressure.

Variables	AN0	AN1	BN	BED	UFED	OSFED	*p*-Value
**Secondary Amenorrhea:**							** *<0.001* **
No	10.5	36.8	37.5	47.4	37.5	50.0	
Yes	89.5	63.2	62.5	10.5	18.8	30.0	
Menopause	0.00	0.00	0.00	42.1	43.8	20.0	
**Food restriction:**							** *<0.001* **
No	0.00	0.00	5.56	36.0	40.0	33.3	
Yes	100	100	94.4	64.0	60.0	66.7	
**Water restriction:**							** *<0.001* **
No	45.5	47.6	66.7	84.0	90.0	91.7	
Yes	54.5	52.4	33.3	16.0	10.0	8.33	
**Fasting:**							** *<0.001* **
No	22.7	20.0	16.7	73.1	80.0	83.3	
Yes	77.3	80.0	83.3	26.9	20.0	16.7	
**Diets pills:**							** *0.002* **
No	100	78.9	94.4	73.1	85.0	83.3	
Yes	0.00	21.1	5.56	26.9	15.0	16.7	
**Binge eating:**							** *<0.001* **
No	84.1	45.0	11.1	0.00	40.0	16.7	
Yes	15.9	55.0	88.9	100	60.0	83.3	
**Vomit:**							** *<0.001* **
No	86.4	15.0	11.1	92.3	84.2	91.7	
Yes	13.6	85.0	88.9	7.69	15.8	8.33	
**Alcohol abuse:**							*0.112*
No	95.3	88.9	75.0	92.3	100	90.9	
Yes	4.65	11.1	25.0	7.69	0.00	9.09	
**Laxatives abuse:**							** *0.008* **
No	93.2	65.0	66.7	92.3	94.7	83.3	
Yes	6.82	35.0	33.3	7.69	5.26	16.7	
**Diuretics abuse:**							** *0.027* **
No	100	84.2	88.2	96.2	100	91.7	
Yes	0.00	15.8	11.8	3.85	0.00	8.33	
**Drugs abuse:**							*0.303*
No	95.5	100	94.4	92.3	100	83.3	
Yes	4.55	0.00	5.56	7.69	0.00	16.7	
**Excessive physical activity:**							** *<0.001* **
No	20.5	10.0	33.3	92.3	90.0	83.3	
Yes	79.5	90.0	66.7	7.7	10.0	16.7	
**Sleep disorder:**							*0.116*
No	20.0	18.2	54.5	50.0	42.9	16.7	
Yes	80.0	81.8	45.5	50.0	57.1	83.3	
**CPAP ventilator:**							** *0.002* **
No	100.0	100.0	100.0	69.2	71.4	100.0	
Yes	0.0	0.0	0.0	30.8	28.6	0.0	

**Table 3 nutrients-15-00406-t003:** Clinical data of ED groups. Summary descriptive table by groups of diagnosis for laboratory variables expressed as median (q1 = first quartile; q3 = third quartile). *p*-values are according to the Kruskal–Wallis test. Statistically significant values are in bold. TSH: thyroid-stimulating hormone, WBCs: white blood cells, RBCs: red blood cells, PLT: platelet, Hb: hemoglobin, Ht: hematocrit, MCV: mean corpuscular volume, fL: femtoliter. Abbreviations: AN0 = restricting anorexia nervosa; AN1 = purging anorexia nervosa; BN = bulimia; BED = binge eating disorders; UFED = unspecified feeding or eating disorder; OSFED = other specified eating disorders.

Variables	AN0	AN1	BN	BED	UFED	OSFED	*p*-Value
**Glycemia, mg/dL**	76.0 [67.8;80.8]	80.0 [72.2;85.0]	80.5 [69.8;84.5]	90.0 [85.0;101]	96.5 [85.5;104]	96.0 [87.0;104]	**<0.001**
**Azotemia, mg/dL**	31.8 [25.0;42.0]	25.5 [20.8;38.2]	28.4 [23.5;31.8]	32.0 [29.2;37.5]	30.0 [27.0;34.5]	34.0 [27.2;37.0]	0.690
**Creatinine, mg/dL**	0.74 [0.62;0.86]	0.71 [0.67;0.82]	0.78 [0.66;0.83]	0.74 [0.62;0.79]	0.77 [0.68;0.90]	0.78 [0.68;0.83]	0.756
**TSH, mU/L**	2.07 [1.42;2.71]	2.00 [1.55;3.04]	1.77 [1.34;2.34]	2.57 [2.05;3.19]	2.71 [1.81;3.26]	2.59 [1.94;4.00]	0.203
**Amylasi, U/L**	68.0 [51.5;101]	84.0 [64.0;104]	69.5 [55.0;79.2]	47.0 [33.0;59.0]	52.0 [37.2;65.2]	65.0 [65.0;65.0]	0.126
**WBCs, ×0^3^/uL**	5.19 [3.90;6.10]	5.14 [4.72;6.88]	6.19 [5.00;7.10]	8.22 [7.12;9.00]	7.18 [6.16;8.34]	6.97 [6.17;8.18]	**<0.001**
**RBCs, ×0^6^/uL**	4.20 [3.90;4.72]	4.43 [4.14;4.47]	4.42 [4.30;4.78]	4.90 [4.73;5.12]	4.82 [4.62;5.24]	4.73 [4.64;5.08]	**<0.001**
**PLT, ×0^3^/uL**	214 [181;238]	221 [200;292]	245 [212;302]	273 [249;312]	260 [221;287]	254 [226;296]	**<0.001**
**Hb, g/dL**	13.2 [12.4;14.1]	13.3 [12.7;13.7]	13.4 [12.1;13.8]	14.1 [13.0;14.7]	13.3 [12.4;14.8]	13.7 [13.2;14.1]	0.270
**Ht, vol%**	38.7 [37.1;41.6]	39.6 [37.5;40.9]	39.5 [38.1;41.2]	43.2 [40.8;44.4]	40.5 [37.8;43.8]	40.5 [39.8;42.9]	**0.001**
**MCV, fL**	90.8 [87.2;94.8]	89.4 [88.4;91.6]	87.6 [85.3;91.7]	87.0 [83.8;89.1]	84.6 [80.6;88.0]	85.8 [82.8;89.9]	**0.002**
**Serum protein electrophoresis, g/dL**	7.30 [6.85;7.75]	7.30 [7.20;7.60]	7.40 [7.00;7.65]	7.00 [6.82;7.58]	7.05 [6.95;7.15]	7.30 [7.30;7.30]	0.814

**Table 4 nutrients-15-00406-t004:** DNA methylation levels of the six CpG sites at SLC6A4 in ED groups and controls. Summary descriptive table by groups of diagnosis expressed as median ([q1 = first quartile; q3 = third quartile) for continuous variable and absolute frequency (% = column percentage) for categorical. Significance values are according to the Kruskal–Wallis test. To show the pairwise differences, we assigned a number to each group: AN0 = (a), AN1 = (b), BN = (c), BED = (d), UFED = (e), OSFED = (f) and, CTR = (*). Statistically significant values are in bold. Abbreviations: AN0 = restricting anorexia nervosa; AN1 = purging anorexia nervosa; BN = bulimia; BED = binge eating disorders; UFED = unspecified feeding or eating disorder; OSFED = other specified eating disorders; CTR = controls.

Variables	AN0 (a)	AN1 (b)	BN (c)	BED (d)	UFED (e)	OSFED (f)	CTR (*)	*p*-Value
**Kyn, µmol/L**	1.35 [1.07;1.57] (d) (e) (f)	1.25 [1.09;1.46] (d) (e) (f)	1.40 [1.06;1.72] (d) (e) (f)	1.97 [1.60;2.35] (*)	1.64 [1.45;2.21] (*)	2.22 [1.90;2.34] (*)	1.37 [1.11;1.58]	** *<0.001* **
**Trp, µmol/L**	50.2 [42.4;53.9] (d) (e) (f)	50.7 [45.8;58.0] (*) (d) (e) (f)	50.6 [46.2;58.4] (*) (d) (e) (f)	41.0 [35.2;47.0] (*)	43.7 [36.0;46.5]	40.6 [36.3;46.1]	45.8 [42.2;49.5]	** *<0.001* **
**Kyn/Trp, * 1000**	25.5 [20.3;31.6] (d) (e) (f)	25.6 [20.8;28.4] (*) (d) (e) (f)	26.1 [23.6;30.1] (d) (e) (f)	42.6 [36.8;54.3] (*)	42.5 [33.8;50.5] (*)	51.8 [42.6;60.6] (*)	28.7 [25.4;34.2]	** *<0.001* **
**SLC6A4 CpG1%**	2.03 [1.69;2.91]	2.24 [1.83;2.69]	3.18 [2.02;4.26]	2.37 [1.89;3.46]	2.44 [1.95;3.59]	2.10 [1.58;2.27]	2.32 [1.83;4.16]	*0.230*
**SLC6A4 CpG2%**	6.04 [5.09;7.40]	7.17 [4.32;9.65]	6.33 [4.99;7.82]	6.10 [4.83;7.94]	5.36 [4.46;7.18]	5.00 [3.66;7.07]	7.31 [5.70;9.22]	*0.270*
**SLC6A4 CpG3%,**	2.40 [1.81;2.77]	2.68 [1.71;3.47]	2.17 [1.86;2.94]	2.42 [2.07;3.14]	2.75 [1.68;3.58]	1.70 [0.00;2.58]	2.91 [2.15;3.66]	*0.121*
**SLC6A4 CpG4%**	3.51 [2.71;4.53]	4.81 [3.63;5.56]	3.97 [3.34;4.59]	4.63 [3.20;5.82]	4.34 [3.61;5.05]	4.33 [3.29;5.13]	4.89 [3.84;5.58]	*0.127*
**SLC6A4 CpG5%**	0.00 [0.00;2.42] (d)	1.75 [0.00;2.28]	0.00 [0.00;2.51]	2.70 [0.91;3.47]	2.18 [1.68;3.05]	0.91 [0.00;2.56]	2.21 [1.58;2.66]	** *0.021* **
**SLC6A4 CpG6%**	0.00 [0.00;2.50] (*)	1.38 [0.00;3.05]	2.21 [0.00;3.55]	2.81 [0.00;3.41]	2.08 [0.00;3.09]	0.00 [0.00;3.13]	2.93 [2.00;3.82]	** *0.025* **
**Mean Methylation%**	2.56 [2.20;3.22] (*)	3.35 [2.37;4.27]	3.25 [2.41;4.03]	3.55 [2.60;4.46]	3.09 [2.17;4.14]	2.78 [2.01;3.05]	3.87 [3.17;4.60]	** *0.008* **

**Table 5 nutrients-15-00406-t005:** Psychopathological data for ED patients. Descriptive statistics by groups of diagnosis for EDI-3 questionnaire expressed as median (q1 = first quartile; q3 = third quartile). The *p*-values result from the Kruskal–Wallis test. Statistically significant values are in bold. DT: drive for thinness, B: bulimia, BD: body dissatisfaction, EDRC: eating disorder risk, LSE: low self-esteem, PA: personal alienation, II: interpersonal insecurity, IA: interpersonal alienation, ID: interoceptive deficits, ED: emotional dysregulation, P: perfectionism, A: ascetism, MF: maturity fears, ECC: eating concerns composite, IC: ineffectiveness composite, IPC: interpersonal problems composite, OC: overcontrol composite, GPM: global psychological maladjustment, ID: interoceptive deficits, APC: affective problems composite, IA: interpersonal alienation, ED: emotional dysregulation, IN: inconsistency, IF: infrequency, NI: negative impression. Abbreviations: AN0 = restricting anorexia nervosa; AN1 = purging anorexia nervosa; BN = bulimia; BED = binge eating disorders; UFED = unspecified feeding or eating disorder; OSFED = other specified eating disorders.

EDI Variables	AN0	AN1	BN	BED	UFED	OSFED	*p-Value*
**DT**	26.0 [22.0;28.0]	28.0 [25.5;28.0]	28.0 [21.0;28.0]	15.0 [10.0;21.0]	17.0 [11.0;21.0]	16.0 [6.00;20.0]	** *<0.001* **
**B**	4.00 [1.00;5.00]	7.00 [4.00;14.0]	21.0 [7.00;22.0]	16.0 [9.00;20.0]	8.00 [4.00;11.0]	10.0 [5.00;15.0]	** *<0.001* **
**BD**	32.0 [24.0;36.0]	36.0 [33.5;38.0]	34.0 [29.0;38.0]	28.0 [22.0;33.0]	28.0 [20.0;32.0]	26.0 [16.0;31.0]	** *0.006* **
**EDCR**	60.0 [44.0;69.0]	70.0 [65.0;75.0]	85.0 [60.0;88.0]	57.0 [47.0;71.0]	54.0 [38.0;60.0]	56.0 [44.0;65.0]	** *0.001* **
**LSE**	19.0 [17.0;21.0]	17.0 [13.0;19.5]	18.0 [16.0;23.0]	10.0 [8.00;16.0]	8.00 [4.00;17.5]	9.00 [6.00;15.0]	** *<0.001* **
**PA**	17.0 [14.0;21.0]	18.0 [10.0;21.5]	21.0 [14.0;24.0]	9.00 [7.00;13.0]	8.00 [4.50;15.0]	9.00 [6.00;14.0]	** *<0.001* **
**II**	17.0 [10.0;19.0]	16.0 [14.5;21.5]	16.0 [12.0;19.0]	11.0 [6.00;16.0]	10.0 [4.50;16.0]	11.0 [6.00;13.0]	** *0.001* **
**IA**	13.0 [11.0;18.0]	12.0 [10.0;17.0]	15.0 [14.0;20.0]	9.00 [7.00;12.0]	9.00 [6.50;11.5]	10.0 [6.00;13.0]	** *0.001* **
**ID**	22.0 [18.0;30.0]	24.0 [18.0;29.5]	23.0 [17.0;25.0]	14.0 [7.00;18.0]	10.0 [6.00;17.0]	14.0 [4.00;18.0]	** *<0.001* **
**ED**	13.0 [10.0;17.0]	15.0 [11.0;16.0]	16.0 [11.2;20.2]	6.00 [3.00;12.0]	6.00 [2.50;10.0]	3.00 [1.00;4.00]	** *<0.001* **
**P**	11.0 [7.00;14.0]	11.0 [8.00;14.5]	11.0 [8.00;19.0]	10.0 [5.00;11.0]	4.00 [2.00;9.50]	5.00 [4.00;12.0]	** *0.006* **
**A**	15.0 [11.0;20.0]	16.0 [12.0;18.5]	14.0 [12.0;21.0]	9.00 [5.00;14.0]	6.50 [4.00;10.0]	9.00 [6.00;13.0]	** *<0.001* **
**MF**	16.0 [14.0;25.0]	16.0 [11.2;21.5]	11.0 [7.00;19.0]	9.00 [7.00;15.0]	13.0 [10.5;16.0]	12.0 [7.00;21.0]	** *0.022* **
**IC**	37.0 [29.0;41.0]	34.0 [27.0;40.5]	40.0 [28.0;46.0]	21.0 [15.0;31.0]	15.0 [9.50;35.5]	19.0 [11.0;28.0]	** *<0.001* **
**IPC**	28.0 [23.0;37.0]	27.0 [25.5;38.5]	31.0 [27.0;36.0]	22.0 [13.0;25.0]	20.0 [12.0;25.5]	23.0 [13.0;25.0]	** *<0.001* **
**APC**	33.0 [25.0;46.5]	39.0 [30.0;45.0]	38.5 [31.5;46.2]	17.0 [12.0;33.0]	14.0 [10.5;27.0]	14.0 [7.00;19.0]	** *<0.001* **
**OC**	27.0 [19.0;33.0]	26.0 [22.0;32.0]	26.0 [22.0;40.0]	16.0 [14.0;26.0]	12.0 [7.50;16.0]	16.0 [11.0;22.0]	** *<0.001* **
**GPM**	148 [114;166]	153 [121;171]	161 [133;166]	96.0 [66.2;114]	73.0 [50.0;115]	62.5 [54.8;112]	** *<0.001* **
**IN**	10.5 [7.75;14.0]	10.5 [8.25;11.8]	5.00 [3.00;10.0]	12.0 [8.00;15.0]	9.00 [8.00;14.5]	8.00 [7.00;12.5]	0.113
**IF**	1.00 [1.00;3.00]	1.00 [0.00;2.00]	1.00 [1.00;3.00]	0.00 [0.00;1.00]	1.00 [0.00;1.00]	0.00 [0.00;1.00]	**0.045**
**NI**	28.0 [13.5;41.5]	25.0 [16.5;37.8]	46.0 [30.0;49.0]	13.0 [8.00;22.0]	11.0 [6.50;25.0]	12.0 [8.00;18.0]	**<0.001**

## Data Availability

All data generated or analyzed during this study are included in this published article.

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
