# Peer review of "SLC6A4 DNA Methylation Levels and Serum Kynurenine/Tryptophan Ratio in Eating Disorders: A Possible Link with Psychopathological Traits?"

_nutrients, 2023, doi:10.3390/nu15020406_

Round 1

Reviewer 1 Report

The manuscript titled “SLC6A4 DNA methylation levels and serum kynurenine/tryptophan ratio in Eating Disorders: a possible link with psychopathological traits?” is well-written and scientifically sound. The authors have studied the epigenetic changes in SLC6A4 gene by determining its methylation levels, correlated it with the metabolites kynurenine/tryptophan (Kyn/Trp) ratio, and psychopathological traits of various eating disorders such as anorexia nervosa (AN), bulimia (BN), binge eating disorders (BED), unspecified feeding or eating disorder (UFED) and other specified eating disorders (OSFED) compared to healthy controls. They found that Kyn was higher in BED, UFED, and OSFED whereas Trp was higher in AN1 and BN compared to these but similar to healthy control. In addition, the CpG methylation levels were low in AN0 as compared to BED and healthy controls. The authors have tried to correlate the molecular pattern (methylation level) with pathological readout (Kyn/Trp) to the psychopathological traits, which is commendable. The manuscript will inspire others in the field to do similar studies, such as linking epigenetic changes to metabolite levels. I only have a few minor comments for the authors, which are as follows:

1.       In Table 4, the authors should include one more row stating the correlation value of methylation with the kyn/Trp. This will help readers to understand the correlation between the two and their significance.

2.       In the Discussion, the authors should discuss the significance of linking epigenetic methylation pattern to the serum Kyn/Trp ratio. They should elaborate on whether these epigenetic changes can help in predicting the classification or severity of various eating disorders in the future; or whether could it be useful to determine the effect of any drug. Such kind of information would assist future studies on the need of the field and what aspect is needed to improve the disease outcome or prediction.

Reviewer 2 Report

Thank you for the opportunity to review this interesting study entitled:  SLC6A4 DNA methylation levels and serum kynurenine/tryptophan ratio in Eating Disorders: a possible link with psycho-pathological traits?   This is a very interesting and current topic.

 There are some topics in the article that should be considered:

Regarding the methodology:  How were the patients selected? is it a  convenience sampling? please clarify .

Concerning the results: I consider the acronyms of the studied groups should be added and clarified in the legends of the tables

I think it is necessary to present the anthropometric and biochemical data of the control group as well. (table 1 and 3 respectively)

The tables must be edited the data in tables 3 and 4 are moved and makes it difficult to understand the results

In the results it is mentioned that a de crease in WBC count is not typically correlated with an increased risk of infection, how was this fact assessed?

I believe that the data on the hematological results should be presented in a separate paragraph.

The phrase Plasma levels of Trp and its derived metabolite Kyn, as well as the Kyn/Trp ratio, were evaluated at the admission of EDs patients, I think it should be in the methodology section.

I think that the data of the concentrations kyn and  Trp as well  kyn/trp of in the different groups should be in the table 3 considering that they are chemical metabolites

The table 3 is not mentioned in the results. All the biochemical markers could be presented together and could eventually include hematological markers

Figure 1 is presented but not mentioned in the text

I think it would be convenient for readers if the tables had titles

Please explain the meaning of the different asterisks in figure 2

Regarding the discussion

How do you interpret the differences between hematological markers and glucose between the different groups? In the same way how do you   explain the relationship of Trp, Kyn and Trp/Kyn with the Binge eating behavior?

How could be interpret the difference between the methylation in specific sites and the general methylation between the studied groups and  having the highest total methylation in the control group? In the discussion these results are summarized again but they are not sufficiently interpreted.

Limitations of the study are not mentioned

Round 2

Reviewer 2 Report

Thanks for submitting the corrected version of the article. I am satisfied with the corrections and propose that the article be accepted in its present form